# Wavelets on Graphs via Deep Learning

**Raif M. Rustamov & Leonidas Guibas**
Computer Science Department, Stanford University
`{rustamov,guibas}@stanford.edu`

## Abstract

An increasing number of applications require processing of signals defined on weighted graphs. While wavelets provide a flexible tool for signal processing in the classical setting of regular domains, the existing graph wavelet constructions are less flexible – they are guided solely by the structure of the underlying graph and do not take directly into consideration the particular class of signals to be processed. This paper introduces a machine learning framework for constructing graph wavelets that can sparsely represent a given class of signals. Our construction uses the lifting scheme, and is based on the observation that the recurrent nature of the lifting scheme gives rise to a structure resembling a deep auto-encoder network. Particular properties that the resulting wavelets must satisfy determine the training objective and the structure of the involved neural networks. The training is unsupervised, and is conducted similarly to the greedy pre-training of a stack of auto-encoders. After training is completed, we obtain a linear wavelet transform that can be applied to any graph signal in time and memory linear in the size of the graph. Improved sparsity of our wavelet transform for the test signals is confirmed via experiments both on synthetic and real data.

## 1 Introduction

Processing of signals on graphs is emerging as a fundamental problem in an increasing number of applications [22]. Indeed, in addition to providing a direct representation of a variety of networks arising in practice, graphs serve as an overarching abstraction for many other types of data. High-dimensional data clouds such as a collection of handwritten digit images, volumetric and connectivity data in medical imaging, laser scanner acquired point clouds and triangle meshes in computer graphics – all can be abstracted using weighted graphs. Given this generality, it is desirable to extend the flexibility of classical tools such as wavelets to the processing of signals defined on weighted graphs.

A number of approaches for constructing wavelets on graphs have been proposed, including, but not limited to the CKWT [7], Haar-like wavelets [24, 10], diffusion wavelets [6], spectral wavelets [12], tree-based wavelets [19], average-interpolating wavelets [21], and separable filterbank wavelets [17]. However, all of these constructions are guided solely by the structure of the underlying graph, and do not take directly into consideration the particular class of signals to be processed. While this information can be incorporated indirectly when building the underlying graph (e.g. [19, 17]), such an approach does not fully exploit the degrees of freedom inherent in wavelet design. In contrast, a variety of signal class specific and adaptive wavelet constructions exist on images and multidimensional regular domains, see [9] and references therein. Bridging this gap is challenging because obtaining graph wavelets, let alone adaptive ones, is complicated by the irregularity of the underlying space. In addition, theoretical guidance for such adaptive constructions is lacking as it remains largely unknown how the properties of the graph wavelet transforms, such as sparsity, relate to the structural properties of graph signals and their underlying graphs [22].

The goal of our work is to provide a machine learning framework for constructing wavelets on weighted graphs that can sparsely represent a given class of signals. Our construction uses the lifting

scheme as applied to the Haar wavelets, and is based on the observation that the update and predict steps of the lifting scheme are similar to the encode and decode steps of an auto-encoder. From this point of view, the recurrent nature of the lifting scheme gives rise to a structure resembling a deep auto-encoder network.

Particular properties that the resulting wavelets must satisfy, such as sparse representation of signals, local support, and vanishing moments, determine the training objective and the structure of the involved neural networks. The goal of achieving sparsity translates into minimizing a sparsity surrogate of the auto-encoder reconstruction error. Vanishing moments and locality can be satisfied by tying the weights of the auto-encoder in a special way and by restricting receptive fields of neurons in a manner that incorporates the structure of the underlying graph. The training is unsupervised, and is conducted similarly to the greedy (pre-)training [13, 14, 2, 20] of a stack of auto-encoders.

The advantages of our construction are three-fold. First, when no training functions are specified by the application, we can impose a smoothness prior and obtain a novel general-purpose wavelet construction on graphs. Second, our wavelets are adaptive to a class of signals and after training we obtain a linear transform; this is in contrast to adapting to the input signal (e.g. by modifying the underlying graph [19, 17]) which effectively renders those transforms non-linear. Third, our construction provides efficient and exact analysis and synthesis operators and results in a critically sampled basis that respects the multiscale structure imposed on the underlying graph.

The paper is organized as follows: in §2 we briefly overview the lifting scheme. Next, in §3 we provide a general overview of our approach, and fill in the details in §4. Finally, we present a number of experiments in §5.

## 2 Lifting scheme

The goal of wavelet design is to obtain a multiresolution [16] of $L^2(G)$ – the set of all functions/signals on graph $G$. Namely, a nested sequence of approximation spaces from coarse to fine of the form $\mathbf{V}_1 \subset \mathbf{V}_2 \subset ... \subset \mathbf{V}_{\ell_{max}} = L^2(G)$ is constructed. Projecting a signal in the spaces $\mathbf{V}_\ell$ provides better and better approximations with increasing level $\ell$. Associated wavelet/detail spaces $\mathbf{W}_\ell$ satisfying $\mathbf{V}_{\ell+1} = \mathbf{V}_\ell \oplus \mathbf{W}_\ell$ are also obtained.

Scaling functions $\{\phi_{\ell,k}\}$ provide a basis for approximation space $\mathbf{V}_\ell$, and similarly wavelet functions $\{\psi_{\ell,k}\}$ for $\mathbf{W}_\ell$. As a result, for any signal $f \in L^2(G)$ on graph and any level $\ell_0 < \ell_{max}$, we have the wavelet decomposition

$$f = \sum_k a_{\ell_0,k}\phi_{\ell_0,k} + \sum_{\ell=\ell_0}^{\ell_{max}-1} \sum_k d_{\ell,k}\psi_{\ell,k}. \tag{1}$$

The coefficients $a_{\ell,k}$ and $d_{\ell,k}$ appearing in this decomposition are called approximation (also, scaling) and detail (also, wavelet) coefficients respectively. For simplicity, we use $a_\ell$ and $d_\ell$ to denote the vectors of all approximation and detail coefficients at level $\ell$.

Our construction of wavelets is based on the lifting scheme [23]. Starting with a given wavelet transform, which in our case is the Haar transform ($HT$), one can obtain lifted wavelets by applying the process illustrated in Figure 1(left) starting with $\ell = \ell_{max} - 1$, $a_{\ell_{max}} = f$ and iterating down until $\ell = 1$. At every level the lifted coefficients $a_\ell$ and $d_\ell$ are computed by augmenting the Haar

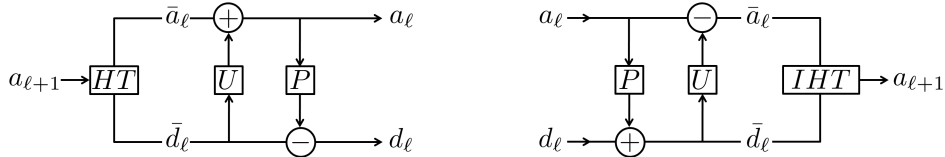

Figure 1: Lifting scheme: one step of forward (left) and backward (right) transform. Here, $a_\ell$ and $d_\ell$ denote the vectors of all approximation and detail coefficients of the lifted transform at level $\ell$. $U$ and $P$ are linear update and predict operators. $HT$ and $IHT$ are the Haar transform and its inverse.

coefficients $\bar{a}_\ell$ and $\bar{d}_\ell$ (of the lifted approximation coefficients $a_{\ell+1}$) as follows

$$
\begin{aligned}
a_\ell &\leftarrow \bar{a}_\ell + U\bar{d}_\ell \\
d_\ell &\leftarrow \bar{d}_\ell - Pa_\ell
\end{aligned}
$$

where update ($U$) and predict ($P$) are linear operators (matrices). Note that in adaptive wavelet designs the update and predict operators will vary from level to level, but for simplicity of notation we do not indicate this explicitly.

This process is always invertible – the backward transform is depicted, with $IHT$ being the inverse Haar transform, in Figure 1(right) and allows obtaining perfect reconstruction of the original signal. While the wavelets and scaling functions are not explicitly computed during either forward or backward transform, it is possible to recover them using the expansion of Eq. (1). For example, to obtain a specific scaling function $\phi_{\ell,k}$, one simply sets all of approximation and detail coefficients to zero, except for $a_{\ell,k} = 1$ and runs the backward transform.

## 3   Approach

For a given class of signals, our objective is to design wavelets that yield approximately sparse expansions in Eq.(1) – i.e. the detail coefficients are mostly small with a tiny fraction of large coefficients. Therefore, we learn the update and predict operators that minimize some sparsity surrogate of the detail (wavelet) coefficients of given training functions $\{f^n\}_{n=1}^{n_{max}}$.

For a fixed multiresolution level $\ell$, and a training function $f^n$, let $\bar{a}_\ell^n$ and $\bar{d}_\ell^n$ be the Haar approximation and detail coefficient vectors of $f^n$ received at level $\ell$ (i.e. applied to $a_{\ell+1}^n$ as in Figure 1(left)). Consider the minimization problem

$$
\{U, P\} = \arg\min_{U,P} \sum_n s(d_\ell^n) = \arg\min_{U,P} \sum_n s(\bar{d}_\ell^n - P(\bar{a}_\ell^n + U\bar{d}_\ell^n)), \tag{2}
$$

where $s$ is some sparse penalty function. This can be seen as optimizing a linear auto-encoder with encoding step given by $\bar{a}_\ell^n + U\bar{d}_\ell^n$, and decoding step given by multiplication with the matrix $P$. Since we would like to obtain a linear wavelet transform, the linearity of the encode and decode steps is of crucial importance. In addition to linearity and the special form of bias terms, our auto-encoders differ from commonly used ones in that *we enforce sparsity on the reconstruction error, rather than the hidden representation* – in our setting, the reconstruction errors correspond to detail coefficients.

The optimization problem of Eq. 2 suffers from a trivial solution: by choosing update matrix to have large norm (e.g. a large coefficient times identity matrix), and predict operator equal to the inverse of update, one can practically cancel the contribution of the bias terms, obtaining almost perfect reconstruction. Trivial solutions are a well-known problem in the context of auto-encoders, and an effective solution is to tie the weights of the encode and decode steps by setting $U = P^t$. This also has the benefit of decreasing the number of parameters to learn. We also follow a similar strategy and tie the weights of update and predict steps, but the specific form of tying is dictated by the wavelet properties and will be discussed in §4.2.

The training is conducted in a manner similar to the greedy pre-training of a stack of auto-encoders [13, 14, 2, 20]. Namely, we first train the the update and predict operators at the finest level: here the input to the lifting step are the original training functions – this corresponds to $\ell = \ell_{max} - 1$ and $\forall n, a_{\ell+1}^n = f^n$ in Figure 1(left). After training of this finest level is completed, we obtain new approximation coefficients $a_\ell^n$ which are passed to the next level as the training functions, and this process is repeated until one reaches the coarsest level.

The use of tied auto-encoders is motivated by their success in deep learning revealing their capability to learn useful features from the data under a variety of circumstances. The choice of the lifting scheme as the backbone of our construction is motivated by several observations. First, every invertible 1D discrete wavelet transform can be factored into lifting steps [8], which makes lifting a universal tool for constructing multiresolutions. Second, lifting scheme is always invertible, and provides exact reconstruction of signals. Third, it affords fast (linear time) and memory efficient (in-place) implementation after the update and predict operators are specified. We choose to apply lifting to Haar wavelets specifically because Haar wavelets are easy to define on any underlying space provided that it can be hierarchically partitioned [24, 10]. Our use of update-first scheme mirrors its

common use for adaptive wavelet constructions in image processing literature, which is motivated by its stability; see [4] for a thorough discussion.

## 4    Construction details

We consider a simple connected weighted graph $G$ with vertex set $V$ of size $N$. A signal on the graph is represented by a vector $f \in \mathbb{R}^N$. Let $W$ be the $N \times N$ edge weight matrix (since there are no self-loops, $W_{ii} = 0$), and let $S$ be the diagonal $N \times N$ matrix of vertex weights; if no vertex weights are given, we set $S_{ii} = \sum_j W_{ij}$. For a graph signal $f$, we define its integral over the graph as a weighted sum, $\int_G f = \sum_i S_{ii} f(i)$. We define the volume of a subset $R$ of vertices of the graph by $Vol(R) = \int_R 1 = \sum_{i \in R} S_{ii}$.

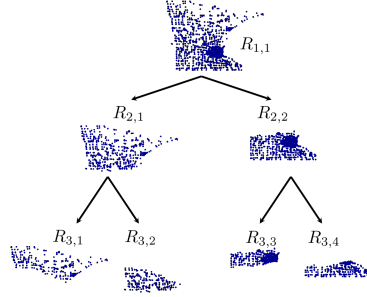

We assume that a hierarchical partitioning (not necessarily dyadic) of the underlying graph into connected regions is provided. We denote the regions at level $\ell = 1, ..., \ell_{max}$ by $R_{\ell,k}$; see the inset where the three coarsest partition levels of a dataset are shown. For each region at levels $\ell = 1, ..., \ell_{max} - 1$, we designate arbitrarily all except one of its children (i.e. regions at level $\ell+1$) as active regions. As will become clear, our wavelet construction yields one approximation coefficient $a_{\ell,k}$ for each region $R_{\ell,k}$, and one detail coefficient $d_{\ell,k}$ for each *active* region $R_{\ell+1,k}$ at level $\ell + 1$. Note that if the partition is not dyadic, at a given level $\ell$ the number of scaling coefficients (equal to number of regions at level $\ell$) will not be the same as the number of detail coefficients (equal to number of active regions at level $\ell + 1$). We collect all of the coefficients at the same level into vectors denoted by $a_\ell$ and $d_\ell$; to keep our notation lightweight, we refrain from using boldface for vectors.

### 4.1    Haar wavelets

Usually, the (unnormalized) Haar approximation and detail coefficients of a signal $f$ are computed as follows. The coefficient $\bar{a}_{\ell,k}$ corresponding to region $R_{\ell,k}$ equals to the average of the function $f$ on that region: $\bar{a}_{\ell,k} = Vol(R_{\ell,k})^{-1} \int_{R_{\ell,k}} f$. The detail coefficient $\bar{d}_{\ell,k}$ corresponding to an *active* region $R_{\ell+1,k}$ is the difference between averages at the region $R_{\ell+1,k}$ and its parent region $R_{\ell,\mathrm{par}(k)}$, namely $\bar{d}_{\ell,k} = \bar{a}_{\ell+1,k} - \bar{a}_{\ell,\mathrm{par}(k)}$. For perfect reconstruction there is no need to keep detail coefficients for inactive regions, because these can be recovered from the scaling coefficient of the parent region and the detail coefficients of the sibling regions.

In our setting, Haar wavelets are a part of the lifting scheme, and so the coefficient vectors $\bar{a}_\ell$ and $\bar{d}_\ell$ at level $\ell$ need to be computed from the augmented coefficient vector $a_{\ell+1}$ at level $\ell + 1$ (c.f. Figure 1(left)). This is equivalent to computing a function's average at a given region from its averages at the children regions. As a result, we obtain the following formula:

$$\bar{a}_{\ell,k} = Vol(R_{\ell,k})^{-1} \sum_{j,\mathrm{par}(j)=k} a_{\ell+1,j} Vol(R_{\ell+1,j}),$$

where the summation is over all the children regions of $R_{\ell,k}$. As before, the detail coefficient corresponding to an active region $R_{\ell+1,k}$ is given by $\bar{d}_{\ell,k} = a_{\ell+1,k} - \bar{a}_{\ell,\mathrm{par}(k)}$. The resulting Haar wavelets are not normalized; when sorting wavelet/scaling coefficients we will multiply coefficients coming from level $\ell$ by $2^{-\ell/2}$.

### 4.2    Auto-encoder setup

The choice of the update and predict operators and their tying scheme is guided by a number of properties that wavelets need to satisfy. We discuss these requirements under separate headings.

**Vanishing moments:**    The wavelets should have vanishing dual and primal moments – two independent conditions due to biorthogonality of our wavelets. In terms of the approximation and detail

coefficients these can be expressed as follows: a) all of the detail coefficients of a constant function should be zero and b) the integral of the approximation at any level of multiresolution should be the same as the integral of the original function.

Since these conditions are already satisfied by the Haar wavelets, we need to ensure that the update and predict operators preserve them. To be more precise, if $a_{\ell+1}$ is a constant vector, then we have for Haar coefficients that $\bar{a}_\ell = c\vec{1}$ and $\bar{d}_\ell = \vec{0}$; here $c$ is some constant and $\vec{1}$ is a column-vector of all ones. To satisfy a) after lifting, we need to ensure that $d_\ell = \bar{d}_\ell - P(\bar{a}_\ell + U\bar{d}_\ell) = -P\bar{a}_\ell = -cP\vec{1} = \vec{0}$. Therefore, the rows of predict operator should sum to zero: $P\vec{1} = \vec{0}$.

To satisfy b), we need to preserve the first order moment at every level $\ell$ by requiring $\sum_k a_{\ell+1,k} Vol(R_{\ell+1,k}) = \sum_k \bar{a}_{\ell,k} Vol(R_{\ell,k}) = \sum_k a_{\ell,k} Vol(R_{\ell,k})$. The first equality is already satisfied (due to the use of Haar wavelets), so we need to constrain our update operator. Introducing the diagonal matrix $A_c$ of the region volumes at level $\ell$, we can write $0 = \sum_k a_{\ell,k} Vol(R_{\ell,k}) - \sum_k \bar{a}_{\ell,k} Vol(R_{\ell,k}) = \sum_k U\bar{d}_\ell Vol(R_{\ell,k}) = \vec{1}^t A_c U \bar{d}_\ell$. Since this should be satisfied for all $\bar{d}_\ell$, we must have $\vec{1}^t A_c U = \vec{0}^t$.

Taking these two requirements into consideration, we impose the following constraints on predict and update weights:

$$\boxed{P\vec{1} = \vec{0} \quad \text{and} \quad U = A_c^{-1} P^t A_f}$$

where $A_f$ is the diagonal matrix of the active region volumes at level $\ell + 1$. It is easy to check that $\vec{1}^t A_c U = \vec{1}^t A_c A_c^{-1} P^t A_f = \vec{1}^t P^t A_f = (P\vec{1})^t A_f = \vec{0}^t A_f = \vec{0}^t$ as required. We have introduced the volume matrix $A_f$ of regions at the finer level to make the update/predict matrices dimensionless (i.e. insensitive to whether the volume is measured in any particular units).

**Locality:**   To make our wavelets and scaling functions localized on the graph, we need to constrain update and predict operators in a way that would disallow distant regions from updating or predicting the approximation/detail coefficients of each other.

Since the update is tied to the predict operator, we can limit ourselves to the latter operator. For a detail coefficient $d_{\ell,k}$ corresponding to the active region $R_{\ell+1,k}$, we only allow predictions that come from the parent region $R_{\ell,\mathrm{par}(k)}$ and the immediate neighbors of this parent region. Two regions of graph are considered neighboring if their union is a connected graph. This can be seen as enforcing a sparsity structure on the matrix $P$ or as limiting the interconnections between the layers of neurons.

As a result of this choice, it is not difficult to see that the resulting scaling functions $\phi_{\ell,k}$ and wavelets $\psi_{\ell,k}$ will be supported in the vicinity of the region $R_{\ell,k}$. Larger supports can be obtained by allowing the use of second and higher order neighbors of the parent for prediction.

### 4.3   Optimization

A variety of ways for optimizing auto-encoders are available, we refer the reader to the recent paper [15] and references therein. In our setting, due to the relatively small size of the training set and sparse inter-connectivity between the layers, an off-the-shelf L-BFGS[1] unconstrained smooth optimization package works very well. In order to make our problem unconstrained, we avoid imposing the equation $P\vec{1} = \vec{0}$ as a hard constraint, but in each row of $P$ (which corresponds to some active region), the weight corresponding to the parent is eliminated. To obtain a smooth objective, we use $L_1$ norm with soft absolute value $s(x) = \sqrt{\epsilon + x^2} \approx |x|$, where we set $\epsilon = 10^{-4}$. The initialization is done by setting all of the weights equal to zero. This is meaningful, because it corresponds to no lifting at all, and would reproduce the original Haar wavelets.

### 4.4   Training functions

When training functions are available we directly use them. However, our construction can be applied even if training functions are not specified. In this case we choose smoothness as our prior, and train the wavelets with a set of smooth functions on the graph – namely, we use scaled eigenvectors of graph Laplacian corresponding to the smallest eigenvalues. More precisely, let $D$ be the diagonal

matrix with entries $D_{ii} = \sum_j W_{ij}$. The graph Laplacian $L$ is defined as $L = S^{-1}(D-W)$. We solve the symmetric generalized eigenvalue problem $(D-W)\xi = \lambda S\xi$ to compute the smallest eigen-pairs $\{\lambda_n, \xi_n\}_{n=0}^{n_{max}}$. We discard the 0-th eigen-pair which corresponds to the constant eigenvector, and use functions $\{\xi_n/\lambda_n\}_{n=1}^{n_{max}}$ as our training set. The inverse scaling by the eigenvalue is included because eigenvectors corresponding to larger eigenvalues are less smooth (cf. [1]), and so should be assigned smaller weights to achieve a smooth prior.

### 4.5 Partitioning

Since our construction is based on improving upon the Haar wavelets, their quality will have an effect on the final wavelets. As proved in [10], the quality of Haar wavelets depends on the quality (balance) of the graph partitioning. From practical standpoint, it is hard to achieve high quality partitions on all types of graphs using a single algorithm. However, for the datasets presented in this paper we find that the following approach based on spectral clustering algorithm of [18] works well. Namely, we first embed the graph vertices into $\mathbb{R}^{n_{max}}$ as follows: $i \rightarrow (\xi_1(i)/\lambda_1, \xi_2(i)/\lambda_2, ..., \xi_{n_{max}}(i)/\lambda_{n_{max}}), \forall i \in V$, where $\{\lambda_n, \xi_n\}_{n=0}^{n_{max}}$ are the eigen-pairs of the Laplacian as in §4.4, and $\xi.(i)$ is the value of the eigenvector at the $i$-th vertex of the graph. To obtain a hierarchical tree of partitions, we start with the graph itself as the root. At every step, a given region (a subset of the vertex set) of graph $G$ is split into two children partitions by running the 2-means clustering algorithm ($k$-means with $k = 2$) on the above embedding restricted to the vertices of the given partition [24]. This process is continued in recursion at every obtained region. This results in a dyadic partitioning except at the finest level $\ell_{max}$.

### 4.6 Graph construction for point clouds

Our problem setup started with a weighted graph and arrived to the Laplacian matrix $L$ in §4.4. It is also possible o reverse this process whereby one starts with the Laplacian matrix $L$ and infers from it the weighted graph. This is a natural way of dealing with point clouds sampled from low-dimensional manifolds, a setting common in manifold learning. There is a number of ways for computing Laplacians on point clouds, see [5]; almost all of them fit into the above form $L = S^{-1}(D-W)$, and so, they can be used to infer a weighted graph that can be plugged into our construction.

## 5 Experiments

Our goal is to experimentally investigate the constructed wavelets for multiscale behavior, meaningful adaptation to training signals, and sparse representation that generalizes to testing signals. For the first two objectives we visualize the scaling functions at different levels $\ell$ because they provide insight about the signal approximation spaces $\mathbf{V}_\ell$. The generalization performance can be deduced from comparison to Haar wavelets, because during training we modify Haar wavelets so as to achieve a sparser representation of training signals.

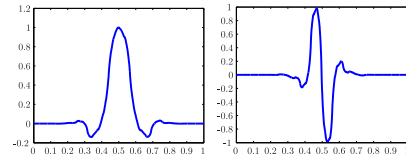

Figure 2: Scaling (left) and wavelet (right) functions on periodic interval.

We start with the case of a periodic interval, which is discretized as a cycle graph; 32 scaled eigenvectors (sines and cosines) are used for training. Figure 2 shows the resulting scaling and wavelet functions at level $\ell = 4$. Up to discretization errors, the wavelets and scaling functions at the same level are shifts of each other – showing that our construction is able to learn shift invariance from training functions.

Figure 3(a) depicts a graph representing the road network of Minnesota, with edges showing the major roads and vertices being their intersections. In our construction we employ unit weights on edges and use 32 scaled eigenvectors of graph Laplacian as training functions. The resulting scaling functions for regions containing the red vertex in Figure 3(a) are shown at different levels in Figure 3(b,c,d,e,f). The function values at graph vertices are color coded from smallest (dark blue) to largest (dark red). Note that the scaling functions are continuous and show multiscale spatial behavior.

To test whether the learned wavelets provide a sparse representation of smooth signals, we synthetically generated 100 continuous functions using the $xy$-coordinates (the coordinates have not been

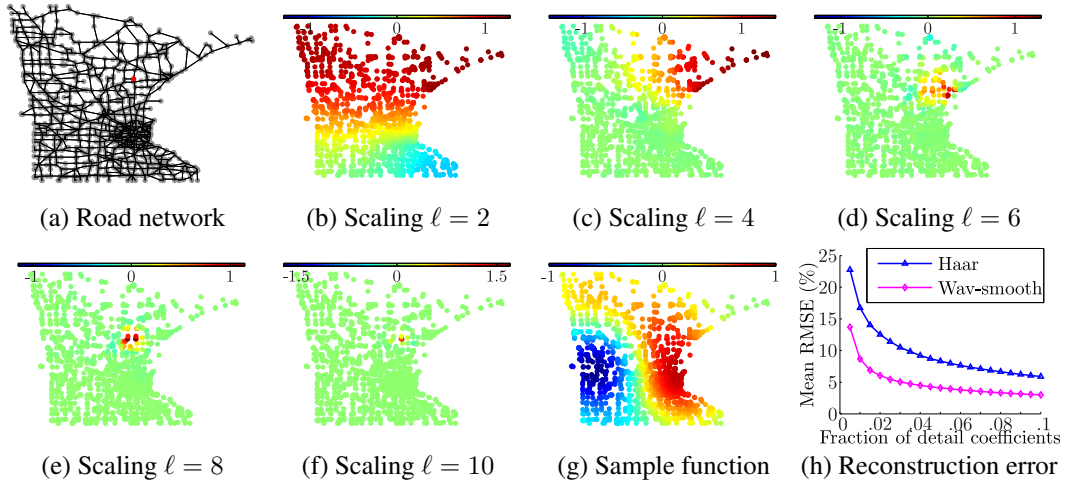

(a) Road network    (b) Scaling $\ell = 2$    (c) Scaling $\ell = 4$    (d) Scaling $\ell = 6$

(e) Scaling $\ell = 8$    (f) Scaling $\ell = 10$    (g) Sample function    (h) Reconstruction error

Figure 3: Our construction trained with smooth prior on the network (a), yields the scaling functions (b,c,d,e,f). A sample continuous function (g) out of 100 total test functions. Better average reconstruction results (h) for our wavelets (Wav-smooth) indicate a good generalization performance.

seen by the algorithm so far) of the vertices; Figure 3(g) shows one of such functions. Figure 3(h) shows the average error of reconstruction from expansion Eq. (1) with $\ell_0 = 1$ by keeping a specified fraction of largest detail coefficients. The improvement over the Haar wavelets shows that our model generalizes well to unseen signals.

Next, we apply our approach to real-world graph signals. We use a dataset of average daily temperature measurements[2] from meteorological stations located on the mainland US. The longitudes and latitudes of stations are treated as coordinates of a point cloud, from which a weighted Laplacian is constructed using [5] with 5-nearest neighbors; the resulting graph is shown in Figure 4(a).

The daily temperature data for the year of 2012 gives us 366 signals on the graph; Figure 4(b) depicts one such signal. We use the signals from the first half of the year to train the wavelets, and test for sparse reconstruction quality on the second half of the year (and vice versa). Figure 4(c,d,e,f,g) depicts some of the scaling functions at a number of levels; note that the depicted scaling function at level $\ell = 2$ captures the rough temperature distribution pattern of the US. The average reconstruction error from a specified fraction of largest detail coefficients is shown in Figure 4(g).

As an application, we employ our wavelets for semi-supervised learning of the temperature distribution for a day from the temperatures at a subset of labeled graph vertices. The sought temperature

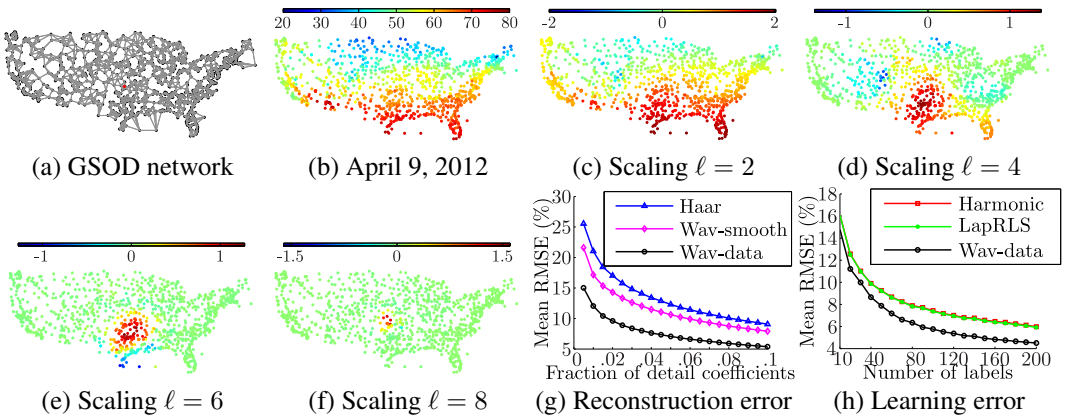

(a) GSOD network    (b) April 9, 2012    (c) Scaling $\ell = 2$    (d) Scaling $\ell = 4$

(e) Scaling $\ell = 6$    (f) Scaling $\ell = 8$    (g) Reconstruction error    (h) Learning error

Figure 4: Our construction on the station network (a) trained with daily temperature data (e.g. (b)), yields the scaling functions (c,d,e,f). Reconstruction results (g) using our wavelets trained on data (Wav-data) and with smooth prior (Wav-smooth). Results of semi-supervised learning (h).

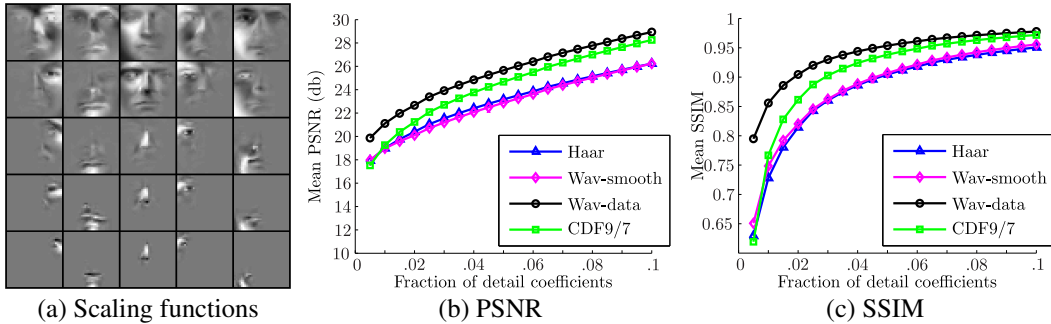

| (a) Scaling functions | (b) PSNR | (c) SSIM |

Figure 5: The scaling functions (a) resulting from training on a face images dataset. These wavelets (Wav-data) provide better sparse reconstruction quality than the CDF9/7 wavelet filterbanks (b,c).

distribution is expanded as in Eq. (1) with $\ell_0 = 1$, and the coefficients are found by solving a least squares problem using temperature values at labeled vertices. Since we expect the detail coefficients to be sparse, we impose a lasso penalty on them; to make the problem smaller, all detail coefficients for levels $\ell \geq 7$ are set to zero. We compare to the Laplacian regularized least squares [1] and harmonic interpolation approach [26]. A hold-out set of 25 random vertices is used to assign all the regularization parameters. The experiment is repeated for each of the days (not used to learn the wavelets) with the number of labeled vertices ranging from 10 to 200. Figure 4(h) shows the errors averaged over all days; our approach achieves lower error rates than the competitors.

Our final example serves two purposes – showing the benefits of our construction in a standard image processing application and better demonstrating the nature of learned scaling functions. Images can be seen as signals on a graph – pixels are the vertices and each pixel is connected to its 8 nearest neighbors. We consider all of the Extended Yale Face Database B [11] images (cropped and down-sampled to $32 \times 32$) as a collection of signals on a single underlying graph. We randomly split the collection into half for training our wavelets, and test their reconstruction quality on the remaining half. Figure 5(a) depicts a number of obtained scaling functions at different levels (the rows correspond to levels $\ell = 4, 5, 6, 7, 8$) in various locations (columns). The scaling functions have a face-like appearance at coarser levels, and capture more detailed facial features at finer levels. Note that the scaling functions show controllable multiscale spatial behavior.

The quality of reconstruction from a sparse set of detail coefficients is plotted in Figure 5(b,c). Here again we consider the expansion of Eq. (1) with $\ell_0 = 1$, and reconstruct using a specified proportion of largest detail coefficients. We also make a comparison to reconstruction using the standard separable CDF 9/7 wavelet filterbanks from bottom-most level; for both of quality metrics, our wavelets trained on data perform better than CDF 9/7. The smoothly trained wavelets do not improve over the Haar wavelets, because the smoothness assumption does not hold for face images.

# 6  Conclusion

We have introduced an approach to constructing wavelets that take into consideration structural properties of both graph signals and their underlying graphs. An interesting direction for future research would be to randomize the graph partitioning process or to use bagging over training functions in order to obtain a family of wavelet constructions on the same graph – leading to over-complete dictionaries like in [25]. One can also introduce multiple lifting steps at each level or even add non-linearities as common with neural networks. Our wavelets are obtained by training a structure similar to a deep neural network; interestingly, the recent work of Mallat and collaborators (e.g. [3]) goes in the other direction and provides a wavelet interpretation of deep neural networks. Therefore, we believe that there are ample opportunities for future work in the interface between wavelets and deep neural networks.

**Acknowledgments:** We thank Jonathan Huang for discussions and especially for his advice regarding the experimental section. The authors acknowledge the support of NSF grants FODAVA 808515 and DMS 1228304, AFOSR grant FA9550-12-1-0372, ONR grant N00014-13-1-0341, a Google research award, and the Max Plack Center for Visual Computing and Communications.

## Footnotes

[1]Mark Schmidt, http://www.di.ens.fr/~mschmidt/Software/minFunc.html

[2]National Climatic Data Center, `ftp://ftp.ncdc.noaa.gov/pub/data/gsod/2012/`

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
