[Reviews · NeurIPS 2013]

Submitted by Assigned_Reviewer_4

The paper proposes an approach for constructing a linear wavelet transform on weighted graphs based on the lifting scheme, which has a number of favourable properties: 1) linear in memory and time with the size of the graph, 2) adaptive to a class of signals, 3) exact analysis and synthesis, i.e. allows for perfect signal reconstruction, 4) efficient training through resemblance with deep auto-encoder networks.

The paper is presented well: it is clearly structured and well written. After a nice overview and introduction, the authors give a detailed derivation of their construction and show in a number of experiments the validity and versatility of their approach.

The proposed approach and wavelet construction builds on previous work but makes a non-trivial contribution to the field of graph-based signal processing by deriving a general-purpose wavelet transform with a number of favourable properties.

The authors make an interesting connection between wavelet construction on graphs and auto-encoder networks. It is likely that this paper will trigger further development in this line of research. It is also likely to serve as a flexible tool in the analysis of signals on graphs.

Additional comments:
* great if the authors could be more precise what sufficient in section 4.5 means? In a general problem how would one determine how many eigenvectors need to be taken into account?
* What is the meaning of the colorbars in Fig. 4 and 5. ?
* In Sec. 4.7 change "It is also possible o" -> "It is also possible to"
Summary: The paper elaborates a non-trivial general-purpose wavelet transform for signals on weighted graphs, which exhibits a number of favourable properties. It makes an interesting connection to auto-encoder networks and is likely to trigger further work along these lines.

Submitted by Assigned_Reviewer_5

This work is aimed to provide interface between the signal processing theory of wavelets and the deep neural network. What is presented is only a small step toward this goal, but is interesting in demonstrating the feasibility of the approach.

It is interesting to see the various connections among wavelet construction and deep auto-encoder.

The detail is difficult to follow, and I hope the presentation can be drastically improved to enhance the readability.
Summary: interesting work to bridge signal processing theory of wavelets and deep learning. But details are difficult to follow, and the presentation should be drastically improved to enhance the readability.

Submitted by Assigned_Reviewer_6

Summary: The authors present a method for constructing wavelets on weighted graphs that can adapt to a class of signals. The approach uses the lifting scheme and by connecting this scheme to a deep auto-encoder network, the authors are able to perform unsupervised training similar to the pre-training of a stack of auto-encoders. The approach is linear in the size of the graph. The authors explore the performance of the wavelet construction approach through application to several data sets.

Quality: The authors provide an elegant approach for taking into account the class of signals when forming wavelets on a graph. Most constructions are based solely on the graph structure, and those few methods that do allow adaptivity based on the signals [19,21] have significant limitations.

The technical development is clearly described, novel, and leads to an efficient algorithm that produces wavelets with desirable properties.

The partitioning approach is described very briefly and there is almost no discussion of how sensitive the approach is to the partitioning method. The spectral clustering algorithm of [20] has some limitations (performance being poor for some kinds of graphs) and it would be nice to see more information about how sensitive the overall wavelet construct is to the partitioning scheme.

For their results on irregular graphs, where I think the construction is of most interest, the authors do not make a comparison to compression or reconstruction using any other type of graph wavelet. Instead, the comparison is to (somewhat dated) learning techniques that are suited to manifold analysis. This constitutes one significant weakness of the paper.

Clarity: The paper is very well written and the development is easy to follow. As discussed above, more detail on some of the

Originality: The authors provide a new method for constructing wavelets on graphs that has the important ability to adapt to the class of functions that the wavelets will be used to represent. The method is highly original.

Significance: The paper represents a useful contribution in the field of wavelets and multiresolution analysis on graphs, a field of growing interest due to the numerous potential applications. I consider that the signal adaptivity, while preserving linearity and efficiency of construction, represents a significant advance.
Summary: The paper makes a significant original contribution, providing an important advance in the field of wavelets on graphs. The lack of a thorough comparison to other wavelet graphs for compression and reconstruction is the major weakness of the paper.
Author Feedback

Author rebuttal: We thank the reviewers for their thoughtful comments and suggestions. We are happy that the reviews generally recognize the interest and novelty of the graph wavelet construction introduced in the paper and share our enthusiasm for further exploration of this area. Below we respond to the main issues raised by the reviewers. Given the opportunity, we will incorporate the necessary modifications into the final version of the paper.


Partitioning:
=========
Since our approach is based on improving upon the Haar wavelets, their quality will have an effect on the final construction. As proved in [Gavish et al], the quality of Haar wavelets depends on the quality of the partitioning; c.f. Theorem 1 and formula (8). From practical standpoint, it is hard to achieve high quality partitions on all types of graphs using a single algorithm. For the graphs used in our paper, we have also run experiments with METIS partitioning as in [Allard et al], and a combination of METIS at coarse levels with spectral partitioning at finer levels, with results similar to what is presented in the paper. Our choice of spectral clustering is motivated by the facts that it has some theoretical guarantees [Szlam] and blends nicely with the rest of exposition. We will be glad to discuss the choice of partitioning in the paper more thoroughly.


Comparison to other wavelets:
=========
We sought to provide a comparison with previous wavelets in a setting with established theory and state of the art algorithms. Our example on the Yale Faces dataset provides such a direct comparison to the state of the art wavelet filters used in JPEG 2000 standard with favorable results. Note that the underlying pixel graph used in our construction is not the standard grid, but each pixel (including the ones on the image boundary) is connected to its 8 nearest neighbors.

If further comparisons with existing graph wavelets are desirable, we are happy to include them in the final version. However, in contrast to our approach, the sparsity of representation has not been directly optimized for in previous graph wavelet constructions; rather it was expected for smooth functions as a consequence of the vanishing moment property. This expectation comes from the setting of the real line, where the number of vanishing moments is the main factor determining the asymptotic decay speed of wavelet coefficients of smooth functions [Mallat, Section 6.1]. However, no generalization of this result is available in graph setting except for Haar wavelets [Gavish et al]. More importantly, to the best of our knowledge, it is not known how to define the notion of higher order vanishing moments for wavelets on graphs in a way that will result in a faster asymptotic decay than that of Haar wavelets. In other words, in terms of sparsity of representation, the Haar wavelets are the state of the art in this nascent field of signal processing on graphs. As stated in [Shuman et al. 2013]: “A major open issue in the field of signal processing on graphs is how to link structural properties of graph signals and their underlying graphs to properties (such as sparsity and localization) of the generalized operators and transform coefficients. Such a theory could inform transform designs, and help identify which transforms may be better suited to which applications.”


As for our comparison to learning techniques, it is motivated by some early attempts of utilizing graph wavelets for semi-supervised learning. [Shuman et al 2011] uses the spectral graph wavelets for semi-supervised learning by including a structured sparsity penalty, yet they find that their prediction performance is sometimes slightly better and sometimes slightly worse than methods based on global smoothness priors. They conclude: “However, this is somewhat disappointing due to the significant additional complexity of the proposed spectral graph wavelet method.” On the other hand, our wavelets achieve improvements over the methods based on global smoothness priors using only a simple sparsity penalty (L1 norm).



Number of eigenvectors to use.
=========
The number of training functions required to robustly train the neural networks depends on the number of parameters; in our case this is related to the number of the neighbors that a region can have at a given level. In the cases discussed in the paper, graphs have a low-dimensional structure, and the number of neighboring partitions is low --- which allows the training to succeed with a small number of training functions. For high dimensional point clouds a larger number (growing with the intrinsic dimension of the manifold) of training functions will be required.


Exposition
=========
We will be glad to follow any specific suggestions for improving the exposition of the paper. We will certainly fix the typos and include an explanation of colorbars in captions of Figs. 4 and 5.


References
=========
[Allard et al.] W.K. Allard, G. Chen, M. Maggioni. Multiscale Geometric Methods for Data Sets II: Geometric Multi-Resolution Analysis. Appl. Comp. Harm. Anal., 32(3): 435-462.

[Gavish et al.] Matan G., Boaz N., Ronald R. C.: Multiscale Wavelets on Trees, Graphs and High Dimensional Data: Theory and Applications to Semi Supervised Learning. ICML 2010: 367-374.

[Mallat] S. Mallat. A Wavelet Tour of Signal Processing, 2nd ed. Academic Press, 1999.

[Shuman et al. 2011] Shuman, D. I.; Faraji, M. J.; Vandergheynst, P.: Semi-Supervised Learning with Spectral Graph Wavelets. SampTA 2011.

[Shuman et al. 2013] D. I. Shuman, S. K. Narang, P. Frossard, A. Ortega, and P. Vandergheynst. The emerging field of signal processing on graphs: Extending high-dimensional data analysis to networks and other irregular domains. IEEE Signal Process. Mag., 30(3):83–98, 2013

[Szlam] A. Szlam. Asymptotic regularity of subdivisions of Euclidean domains by iterated PCA and iterated 2-means. Appl. Comp. Harm. Anal., 27(3): 342-350, 2009.